# Oxidative Stress in Male Infertility: Causes, Effects in Assisted Reproductive Techniques, and Protective Support of Antioxidants

**DOI:** 10.3390/biology9040077

**Published:** 2020-04-10

**Authors:** Jordi Ribas-Maynou, Marc Yeste

**Affiliations:** 1Biotechnology of Animal and Human Reproduction (TechnoSperm), Institute of Food and Agricultural Technology, University of Girona, E-17003 Girona, Spain; 2Unit of Cell Biology, Department of Biology, Faculty of Sciences, University of Girona, E-17003 Girona, Spain

**Keywords:** sperm, oxidative stress, male infertility, reactive oxygen species, antioxidants, assisted reproduction, pregnancy, DNA damage, 8-OHdG

## Abstract

The spermatozoon is a highly specialized cell, whose main function is the transport of the intact male genetic material into the oocyte. During its formation and transit throughout male and female reproductive tracts, sperm cells are internally and externally surrounded by reactive oxygen species (ROS), which are produced from both endogenous and exogenous sources. While low amounts of ROS are known to be necessary for crucial physiological sperm processes, such as acrosome reaction and sperm–oocyte interaction, high levels of those species underlie misbalanced antioxidant-oxidant molecules, generating oxidative stress (OS), which is one of the most damaging factors that affect sperm function and lower male fertility potential. The present work starts by reviewing the different sources of oxidative stress that affect sperm cells, continues by summarizing the detrimental effects of OS on the male germline, and discusses previous studies addressing the consequences of these detrimental effects on natural pregnancy and assisted reproductive techniques effectiveness. The last section is focused on how antioxidants can counteract the effects of ROS and how sperm fertilizing ability may benefit from these agents.

## 1. Introduction

Human infertility is recognized as a disease by the World Health Organization (WHO) and has nowadays an estimated prevalence of 7–15%, with 50–80 million people being affected around the world [1,2]. These high ranks of incidence exemplify the differences between developed and developing countries, the latter being the ones with higher rates, a fact that has been associated to lower familiar income and less health-care accessibility [3,4]. Despite these worrying numbers, mounting evidence from surveys and comparative studies suggest that infertility grows year over year [4,5]. In this scenario, and despite the high relevance given to the female factor in the nineties, it is now well accepted that dysfunctions can also affect spermatozoa, so that a male factor is thought to be present in half of the infertile couples, being the only or main cause of infertility in about 20–30% of cases [6,7,8]. For this reason, research focused on the male factor has been increasing over the last decades, revealing new knowledge about causes and treatments of infertility. Although significant advances on understanding the etiology of male infertility have been made, little is still known about the underlying physiopathology in a high number of particular cases. In most couples, attributing a single cause to infertility is difficult since there is usually a multifactorial, sometimes unknown, pathology. Recent research has come up with different indicators and biomarkers that are able to explain male infertility, such as some specific related genes, concrete epigenetic profiles, and several mRNAs that are responsible for poor sperm quality and/or embryo development [9,10,11]. On the other hand, anatomical affectations such as vas deferens agenesia, hormonal issues causing azoospermia, or varicocele are also factors that help diagnose infertility in the male [12,13]. Moreover, increasing male age, environmental pollution, diet, heat stress, and obesity are also known to be causes for a reduction of sperm quality. In these latter ones, however, it is difficult to establish their precise incidence in the physiopathology of a single male, despite the efforts in defining ranks or risk areas for each parameter. In a scenario where a human male is exposed to some of these exogenous affectations, an increase in oxidative stress is usually observed [14]. An excess of reactive oxygen species (ROS) can cause unbalance with the antioxidant capacity, reaching pathological levels and originating fertility issues. Infertility caused by oxidative stress has been widely studied for years, and multiple studies analyzing its clinical effect have been performed, in relation to both natural fertility and assisted reproductive techniques. Therefore, the aims of the present review are (a) to summarize the causes and consequences of oxidative stress in male fertility; (b) to provide an update about the clinical association between oxidative stress and fertility rates; and (c) to discuss the role of antioxidants in ameliorating fertility outcomes.

## 2. Sources of Oxidative Stress with Potential Effects to the Male Germline

Oxidative stress is related to the disbalance of oxidant molecules in cells. Reactive oxygen species refer to oxygen-derivate molecules that are highly reactive due to their free electrons or radicals. These group of molecules, which have a half-life of nanoseconds, include superoxide (·O_2_^−^), hydrogen peroxide (H_2_O_2_), proxyl radical (·R OO), or hydroxyl (·OH^−^). Less common but also present in sperm cells are the reactive nitrogen species (RNS), which include nitric oxide (·NO), dinitrogen trioxide (N_2_O_3_), and peroxinitrite (ONOO-). The presence of oxidative stress affecting sperm function has been consistently reported since the late forties and early fifties, when different authors pointed out that hydrogen peroxide exerted a detrimental impact on sperm [15]. Nowadays, this damage is irrefutable and, over the last decades, multiple investigations have been oriented towards identifying the possible sources of ROS in order to understand their action mechanism and their consequences. This knowledge has allowed the discovery of new treatment options for human patients, thus improving their reproductive chances. Regarding the male germline, reactive oxygen species can be produced either endogenous or exogenous. From here on, we review these two ways of producing ROS that, despite being necessary at low levels, contribute to a pathological state at high levels. 

### 2.1. Endogenous Sources of ROS

One should note that the spermatozoon itself is a source of ROS due to its metabolic activity, and that other immature sperm cells present in the semen are also important sources of free radicals. While disbalances of ROS have disruptive effects that will be described below in other sections, low levels of ROS are known to be necessary for the sperm cell to perform natural functions. For instance, nitric oxide and hydrogen peroxide are necessary compounds to achieve capacitation, enabling acrosome reaction, which is controlled by reactive oxygen species [16,17]. Moreover, sperm hyperactivation or its interaction with the oocyte is mediated by ROS, thereby being also essential for achieving their fertilizing ability [18,19] (Figure 1). For these reasons, after spermatogenesis and epididymal maturation sperm cells have low levels of oxidative radicals.

Endogenous free radicals can be generated as a by-product of cell metabolism, or by enzymatic activity. Sperm metabolism is a major source of ROS (Figure 1). The activity of mitochondria, which are located in the mid-piece, is, together with glycolysis, essential for sperm motility and capacitation. These mitochondria generate ATP through respiratory electron chain and oxidative phosphorylation, which are based on transferring electrons from inner mitochondrial membrane complexes to oxygen and on pumping of protons to the intermembrane space. These protons are ultimately used to synthesize ATP via complex V (ATP synthase). In this scenario, ROS are by-products of the electron chain activity and are especially generated at complexes I and III, which mainly release superoxide and hydroxyl radicals into the matrix and the intermembrane space [20]. These free radicals are then converted into hydrogen peroxide by superoxide dismutase.

During spermatogenesis, release of sperm with retained cytoplasm usually occurs. Retention of cytoplasm leads to non-functional and immature sperm which retain glucose-6-phosphate dehydrogenase (G6PDH) that enables the production of intracellular β-nicotinamide adenine dinucleotide phosphate (NADPH). This intracellular NADPH is processed via NADPH oxidase which regenerates NADPH to NADP converting O_2_ to superoxide, which is converted to hydrogen peroxide at the intermembrane space by the superoxide dismutase [21,22]. 

The hydrogen peroxide generated through these two different ways is scavenged by glutatione peroxidase or glutathione-s-reductase, which use reduced glutathione (GSH) as an electrone donor. Reduced glutathione is maintained by an ATP-consuming process, through glutathione synthetase (GSS) or glutathione reductase (GSR), which transforms oxidized glutathione (GSSG) into reduced glutathione in a NADPH dependent way [23,24].

### 2.2. Exogenous Sources of ROS

#### 2.2.1. Varicocele

Varicocele is a dilatation of the pampiniform plexus of the spermatic cord that is considered the most common correctable cause of male infertility. The incidence of varicocele in the general population is about 15%, and different studies have determined that its incidence among infertile men is about 35–44%, these figures increasing up to 45–81% in the case of secondary infertility [12]. The dilatation of varicose veins coupled to insufficient venous valves cause a blood reflux and an increase in the blood pressure to the vein wall; this increases testis temperature, which usually has to be 2 °C lower than that of the body. These two varicocele consequences lead to an increase of reactive oxygen species and a reduction of antioxidant capacity [25] (Figure 1). Varicoceles are divided into different grades (subclinical and clinical, with grades 1, 2, or 3) according to their clinical features. It is known that their effect is higher as grade increases, and several studies have pointed out that varicocelectomy, the most common procedure for varicocele correction, is an effective method to reduce sperm OS and sperm DNA damage in clinical varicoceles [26,27]. 

#### 2.2.2. Infections and Leucocytospermia

Immune response against infections cause inflammation of tissues to promote leucocyte infiltration. Leukocytes are an important source of oxidative stress, and it has been described that a single leukocyte produces 1000 times more ROS than a single spermatozoon, via increasing NADPH production [28] (Figure 1). Although every ejaculate contains a certain number of leukocytes, leucocytospermia consists of an increase in that number and usually results from infections [29]. While the increase in ROS has been reported to affect both leukospermic and non-leukospermic patients, it is apparent that the extent at which patients with augmented leukocyte counts suffer from higher oxidative stress is higher [30]. This increment has been demonstrated to be associated to a detrimental impact on different conventional semen parameters, such as motility, morphology, and concentration [31,32,33].

#### 2.2.3. Alcohol and Tobacco

It is well known that consumption of alcohol, tobacco, and different recreational drugs contribute to serious, negative effects on the organism. Regarding alcohol, a study in rats concluded that continued alcohol intake causes a decrease in testicular reduced glutathione concentration, a decrease in testicular superoxide dismutase activity, an increase in testicular malondialdehyde concentration, and an increase in sperm DNA damage. In addition, fertility rates of male rats ingesting alcohol were demonstrated to be lower than those of the control group [34]. Other studies reported similar results, describing detrimental effects on mitochondria, with a significant increase in ROS generation, and observing epigenetic modifications in the germline [35,36]. In addition, different studies have shown that chronic ethanol intake has been related to a decrease of cell proliferation in testes; to the induction of testicular apoptosis, increasing the Fas ligand and upregulating p53 gene expression; and to epididymal damage [37,38,39]. A deregulation of the apoptotic response at testicular level has been described as an issue caused by oxidative stress, and named as abortive apoptosis [40,41]. Moreover, autophagy can be also activated as a protective role to cooperate with apoptosis during spermatogenesis [42,43] 

On the other hand, tobacco smokers are exposed to thousands of chemicals, which are demonstrated to be carcinogenic and the cause of several diseases that may lead to death. Most of these chemicals are demonstrated to increase free radicals and ROS coupled to a reduction of antioxidant activity, which leads to a higher rate of sperm DNA fragmentation and loss of sperm motility [44,45] (Figure 1). 

#### 2.2.4. Physical Exercise and Heat Stress

Exercise is beneficial for different aspects of health. In the case of reproductive physiology, while improved fertility potential has been identified in animals exposed to regular exercise [46] and sedentary men have been reported to present worse sperm quality [47,48], other studies have shown that exercise has no improving effect on conventional spermiogram parameters [49]. In addition, intense physical exercise or cycling is considered to be detrimental for fertility, and both an increase in oxidative stress and a reduction in sperm motility were observed in studies focusing on these types of exercise [50,51]. This increment in oxidative stress and the reduction in sperm quality could also be related to an increase in testicular temperature (Figure 1). Scrotal temperature is known to be 2 °C lower than that the body core temperature, and increases in this value have been shown to impair sperm quality. In fact, it has been reported that every 1 °C of increase correlates to a 14% drop in sperm production [52]. In the same way, some reports have shown that habits such as prolonged car sitting, taking regular sauna bath, or wearing tight fitting underwear can have an impact on testis heat stress, reducing sperm quality [53]. 

#### 2.2.5. Radiations and Pollution

Radiations can be divided into ionizing and non-ionizing. Non-ionizing radiation is the most used by human beings, and one can find in this category from cell phones, which use extremely low frequency, to microwave ovens and radars, which are in the radio frequency range [54]. The effect of non-ionizing radiations produced by mobile phones, microwaves or WIFI devices to sperm oxidative stress and its fertilizing ability is a topic of increasing interest among the scientific community. Different multiple in vitro and in vivo studies aiming at addressing this issue by testing sperm from animal models and human beings have shown that mobile phone’s radiation causes an increase in reactive oxygen species that induces lipid peroxidation and a decline in the antioxidant capacity, induced, amongst others, by the decrease in reduced glutathione levels [55,56,57,58]. Moreover, other studies have associated laptop computer WIFI to increases of DNA damage and decreases of sperm parameters, such as motility, count, and morphology [59,60]. In summary, although non-ionizing radiations are not able to cause DNA alterations directly, they have an indirect potential of affecting fertility through increasing pro-oxidant molecules (Figure 1).

X-rays, γ-rays, and α-particles are ionizing radiations, which are rather more dangerous than the non-ionizing ones at different health levels. Exposure of cells to ionizing radiations increase ROS generation and induce their senescence [61]. Radiation induces direct DNA breaks and potentially affects proteins and membranes through increased ROS levels (Figure 1). In fact, serious health problems, such as different types of cancer, can arise from the exposure to ionizing radiation; in this context, it is worth noting that, due to their lack of antioxidant defense, sperm cells are especially vulnerable [62]. 

Environmental pollution has been found to increase ROS generation and lead to a reduction in sperm quality. For instance, on the one hand and according to studies analyzing semen from human males exposed to traffic pollution, car smoke pollutants potentially reduce men fertility [63] by affecting membrane lipids, generating DNA damage, and even changing expression patterns of proteins involved in spermatogenesis [64,65,66]. On the other hand, continuous exposure to phthalate derivatives have been correlated to increases in reactive oxygen species and decreases in enzymatic and non-enzymatic antioxidants in animal models [67], and in the seminal plasma of infertile patients involved in assisted reproductive programs [68]. While these are only two examples of how air pollutants may affect men fertility, strong evidence is still required to support the urgent need of reducing environmental toxicants to improve reproductive efficiency [69].

## 3. Effects of Oxidative Stress to Sperm Components

All the aforementioned causes have been related to increases in intracellular ROS levels, and the particular and specific organization of the sperm cell makes it especially vulnerable to these increments. The sperm cell does not present a DNA repair machinery since it is transcriptionally silent and possesses either no or poor translational activity [70]. Moreover, at the end of spermiogenesis, testicular spermatozoa lose their cytoplasm, leaving their DNA highly vulnerable to ROS. Therefore, neither cytosolic antioxidant enzymes, such as glutathione peroxidase (GPX), catalase (CAT) and superoxide dismutase (SOD), which are the most common cytosolic ROS-scavengers, nor the Base Excision Repair pathway (BER) are present in sperm cells [70]. Regarding the BER pathway, only an active form of 8-oxoguanine DNA glycosylase (OGG1) enzyme, which excises the oxidized base 8-hydroxy-2′-deoxyguanosine (8OHdG), is left [23]. Since downstream enzymes involved in the BER are missing, the process is stopped after OGG1-excission. For this reason, one can usually find ejaculated sperm with damaged DNA. However, evolution has endowed the oocyte with enzymes devoted to paternal DNA repair after fertilization [71], which unequivocally underpins the collaboration between male and female gametes during the first stages of embryo development. 

The aforementioned exposure of sperm cells to different sources of ROS, which are free to travel across the membrane and between mid-piece and the nucleus, and their inability to scavenge ROS make sperm cells very susceptible to damage. This fact leads to different affectation grades, especially targeting their membranes and nuclei, inducing lipid peroxidation, protein alterations, and DNA damage. 

### 3.1. Lipid Peroxidation

Sperm plasma membrane has a particular susceptibility to oxidative damage, as it contains high amounts of polyunsaturated fatty acids, which present multiple double bonds, the decohexaenoic acid (DHA) being the most representative one [70,72,73]. These membrane molecules are reactive to oxygen radicals, producing highly reactive lipid aldehydes, like malondialdehyde, that have the potential of causing DNA damage and modify proteins. It has been described that these aldehydes inhibit some antioxidant enzymes like G6PDH which, in turn, reduce the activity of glutathione peroxidase [18,23]. Lipid peroxidation alters membrane fluidity and permeability, which results in sperm motility loss and in a reduced sperm ability to interact with the oocyte [23].

### 3.2. Protein Modifications

Proteins are the target of redox reactions, which can activate or inactivate their functionality. Depending on the type of ROS, proteins can be altered through thiol oxidation, tyrosine nitration, sulfonation, or glutathionylation. Thiol oxidation is the most common protein modification that targets glycolytic and Krebs cycle enzymes, which leads to a reduction in the efficiency of ATP production; targets α-tubulin, which impairs microtubule polymerization; and targets protamines, which affect chromatin remodeling during spermiogenesis [74].

### 3.3. Sperm DNA Damage

Sperm DNA fragmentation is the other major effect directly or indirectly caused by oxidative stress. In contact to sperm DNA, ROS cause base modifications, such as 8-OH-guanine and 8-OH-2′-deoxyguanosine which, after being excised by OGG1 enzyme, generate an abasic site that gives place to a single-strand DNA break. Since, as aforementioned, spermatozoa are devoid of enzymes allowing the continuation of the BER pathway and the silencing of sperm chromatin, this break site cannot be repaired by the sperm cell itself and, thus, sperm with fragmented DNA are observed in the ejaculate [70]. 

Oxidative stress can also promote nuclear decondensation, increasing the DNA susceptibility to be damaged by free radicals, which will thus have easier access to the entire sperm genome. Related with this, incubation of spermatozoa from different animals with H_2_O_2_ indicates that DNA breaks occur in a dose-dependent manner; at a final concentration of 10 mM, H_2_O_2_ affects around 60% of sperm DNA [75]. In this context, however, it is worth mentioning that detecting these DNA breaks and its type heavily relies upon the sensitivity of the technique used to evaluate sperm DNA fragmentation (i.e., SCSA, SCDt, TUNEL, neutral Comet, alkaline Comet, etc.), which leads to different correlations to clinical outcomes [76].

## 4. Effects of Oxidative Stress on Fertility Treatments

### 4.1. Natural Pregnancy

A disbalance of oxidative stress is usually the cause of different alterations heading, in the end, to DNA fragmentation, protein and lipid modifications, and affectations in sperm count, motility, and morphology. Although it is difficult to establish which percentages of infertile patients have an underlying mechanism of oxidative stress, it is known that it could be present as a factor in asthenozoospermic, teratozospermic, and oligozoospermic patients who present reproductive issues [77,78]. All these parameters have been associated in different studies with a reduction in natural pregnancy, both in animal [79,80] and human studies [76,81,82], supporting the evidence that oxidative stress leads to a reduction in natural fertility rates [77,83]. 

### 4.2. Assisted Reproduction Techniques

In a scenario where natural pregnancies are impaired for at least 12 months, assisted reproduction techniques (ART) are available treatments for infertile couples, and include intrauterine insemination (IUI), in vitro fertilization (IVF), and intracytoplasmic sperm injection (ICSI). While the contribution of the male factor infertility to the success rates of different ART has been the source of much debate during the last decade, the following two subsections summarize our current knowledge about the effects of oxidative stress on the effectiveness of those treatments. 

#### 4.2.1. Intrauterine Insemination

Three systematic reviews and meta-analyses related to the topic have been identified, with the general conclusion that oxidative stress and sperm DNA damage have negative effects on pregnancy rates after IUI. In the first one, the authors report different studies supporting that sperm DNA damage is related to lower pregnancy rates after IUI, since altered sperm DNA can lead to a decrease in pregnancy rates from 24% to 3% [84]. In the second meta-analysis, similar conclusions were reached from the selected studies included in the systematic review, emphasizing that the extent of the impact of sperm DNA damage on pregnancy rates after IUI is higher when male age is over 30 years old [85]. Finally, a meta-analysis by Belgian researchers including 940 IUI cycles revealed that low DNA damage increases by 3.15 the chances of achieving clinical pregnancy after IUI [86].

#### 4.2.2. IVF and ICSI

Different meta-analyses aimed at addressing the relationship between sperm DNA damage and IVF/ICSI success rates have been conducted. Regarding IVF, separate studies have concluded that sperm DNA damage may have an important influence in preventing clinical pregnancy, increasing miscarriage, or even decreasing embryo quality [84,87,88,89]. 

As far as ICSI is concerned, data are less clear since whereas some works have reported that sperm DNA damage influences treatment outcomes, such as pregnancy rates and embryo quality [88,89], other meta-analyses have concluded that sperm DNA fragmentation has no influence on ICSI performance [84,90,91]. 

Inconsistent results from studies analyzing combined IVF and ICSI data are found in the literature, with some meta-analyses supporting that sperm DNA damage causes negative effects on ART [88] and others showing that there is no enough evidence to sustain that assertion [92,93]. These contrasting conclusions may not only be due to the inclusion of different techniques but also to the heterogeneity of different laboratories and populations.

## 5. Protection against Oxidative Damage and Designed Treatments for Fertility Improvement

The principal mission of the sperm cell is to transport and deliver the intact paternal DNA into the oocyte. For this reason, sperm DNA has evolved to be as much protected as possible from both internal and external sources of damage. This evolution has led to an almost inert state, where histones are replaced in most parts of the genome by protamines, which remain organized in toroids condensing about 50 kilobases of DNA. These toroids are linked to each other by toroid linker regions that are condensed with histones [94]. While the role of histone-condensed regions is currently under debate, recent studies suggest that they could include gene promoter regions, such as CpG-rich islands and satellite repeats [95,96]. In addition, telomeres have also been described to retain a high percentage of histones [97]. Whilst protaminated toroidal regions have been shown to be resilient against nuclease activity and histone-condensed regions seem to be more susceptible to DNA damage, incubation of sperm with H_2_O_2_ indicates that oxidative damage can induce DNA breaks in both regions [98].

### 5.1. Seminal Plasma

Due to the loss of the most part of the cytoplasm and the lack of transcription and translation potential, the sperm cell is devoid or has very low levels of antioxidant enzymes in the cytosol. In order to avoid the disbalance of ROS generating oxidative stress, the low antioxidant activity of mature spermatozoa has to be compensated by the antioxidant capacity of seminal plasma [99]. As a result of this atoning role, seminal plasma has evolved as one of the most known antioxidant fluids, estimating that its antioxidant power is more than 10 times higher than that of the blood [100]. This includes both antioxidant enzymes, such as CAT, SOD, glutathione-s-transferase (GST), GPX and GSR, and small free-radical scavengers, such as vitamin C, polyphenols, carotenoids, and coenzyme Q10 [100,101,102]. Taking into account both pro-oxidant and antioxidant activities, it seems reasonable to suggest that measuring redox balance may provide a broader picture on the relationship between sperm oxidative stress and male infertility [70], this balance being altered either by the increase of ROS or by the decrease of antioxidant activity. 

### 5.2. Varicocelectomy

As mentioned before in Section 2, varicocele is a pathology that induces an increase of oxidative stress in the male reproductive system. The surgical treatment of varicocele has been demonstrated to be an effective method to remove the alterations caused by the affected veins, as blocking blood reflux prevents temperature increases. The effects observed after the recovery from surgery include a decrease in reactive oxygen species and an increase in total antioxidant capacity, which comes from augmenting the levels of antioxidant enzymes and molecules [27,103,104]. As a consequence, DNA damage is reduced after varicocelectomy [105]. In addition, a placebo-controlled, double-blind trial showed that co-treatment with melatonin as an antioxidant may improve the results obtained following surgery [106].

### 5.3. Antioxidants

Since not only does an increase in oxidative stress reduce sperm quality but also the likelihood of achieving pregnancy (either natural or through ART), the use of exogenous antioxidants has been proposed to balance the ROS:antioxidant ratio and increase the sperm quality. In the literature, several non-enzymatic antioxidants, such as arginine, carnitine, carotenoids, coenzyme Q10, cysteine, reduced glutathione, micronutrients like selenium or zinc, vitamin E, vitamin C, myo-inositol, or resveratrol, have been reported to be utilized to treat different diseases [107]. In the case of ART, exogenous antioxidants have been tested for decades, and several works point out to a positive contribution of these supplements on sperm count, motility, and morphology [108,109,110]. Other studies have also proved that antioxidants are useful to ameliorate lipid peroxidation, reduce DNA base modifications (like 8-OHdG) and fragmentation, and increase total antioxidant capacity [111,112,113,114]. Finally, previous research has also demonstrated that adding cryopreserved sperm with antioxidants increases their quality [115,116,117]. 

#### 5.3.1. Use of Antioxidants for Natural Pregnancy

Given that sperm quality has been associated to higher success in achieving natural pregnancy, one could suggest that improving sperm function and survival through antioxidant supplementation should cause a positive effect on fertilization success (Figure 2). However, at present, only few studies with randomized and placebo controls prove that association. The latest review of the Cochrane Library including high quality studies concludes that the antioxidant intake by male patients is associated to an increase in live birth rates from 12% to 14–26%, representing between 1.2- and 2.1-fold increase. The same meta-analysis included eleven studies analyzing natural pregnancy and showed that antioxidant supplementation increases pregnancy rates from 7% to 12–26%, representing between 1.7- and 3.7-fold increase [107]. While these results seem to be promising, the authors of that study warned about the increase in miscarriage rates (from 2% to 13%) following antioxidant supplementation. This detrimental effect, however, was concluded from few studies (only three) meeting the quality criteria (randomized, prospective, and double-blinded) [107].

#### 5.3.2. Use of Antioxidants for Intrauterine Insemination

The use of antioxidants with patients undergoing ART is usual. Nevertheless, the potential benefits have been the subject of controversy in the field. Although it seems a common finding that antioxidants have a positive effect on sperm quality parameters, the very low amount of high quality studies foster the controversy on whether or not their use increases male fertility in the case of IUI (Figure 2). For instance, a randomized controlled trial using *N*–acelylcysteine and a randomized double-blind trial using astaxanthin as antioxidants in couples performing IUI observed an increase in pregnancy rates compared to controls [118,119]. In contrast, a multicenter, double-blind, randomized, placebo-controlled trial published in 2020, and conducted in the United States between 2015 and 2018 concluded that an antioxidant formulation containing vitamins C, E, selenium, L-Carnitine, zinc, folic acid, and lycopene had no effect on sperm quality parameters, pregnancy, or live birth rates [120]. The limitation of these two studies, however, was the low number of patients.

#### 5.3.3. Use of Antioxidants for IVF/ICSI

While randomized and controlled trials are much warranted in this realm, few studies have analyzed the effects of antioxidants on pregnancy rates after IVF or ICSI, usually with small sample sizes, thus reinforcing the need of large studies. A prospective, randomized, double-blind placebo-controlled study performed in Australia between 2004 and 2006 conducted with 60 patients found that a mix of commercial antioxidants (which included lycopene, Vitamins E and C, zinc, selenium, folate, and garlic) taken orally during three months prior to IVF increased both pregnancy and implantation rates [121]. A non-controlled study showed that vitamin E intake during one or three months increased fertilization rates per cycle after IVF regardless of intake time [122]. Furthermore, another non-controlled study conducted with 38 patients presenting a previous failed ICSI cycle and high sperm DNA fragmentation reported that administrating vitamins C and E orally for two months prior to the second ICSI attempt improved clinical pregnancy and implantation rates compared to the first ICSI attempt [123].

## 6. Conclusions

Despite the vital function of ROS during fertilization, an increase in those species causes oxidative stress and detrimentally affects sperm function. The origin of this increase arises from different endogenous and exogenous sources, such as sperm metabolism or infections, respectively. It is well known that ROS cause a reduction in sperm count and motility, protein alterations, lipid peroxidation, and DNA fragmentation, amongst others. These affectations to sperm quality lead to lower natural pregnancy rates and decline IUI and IVF success. Regarding ICSI, more studies are needed to elucidate which the actual impact of high ROS levels in sperm is. Finally, while using exogenous antioxidants as a method to counteract the adverse effects of oxidative stress has been proven to be effective for natural pregnancy, randomized, double-blind, prospective, and placebo-controlled studies are still needed to demonstrate their effectiveness in assisted reproductive techniques. 

## Figures and Tables

**Figure 1 biology-09-00077-f001:**
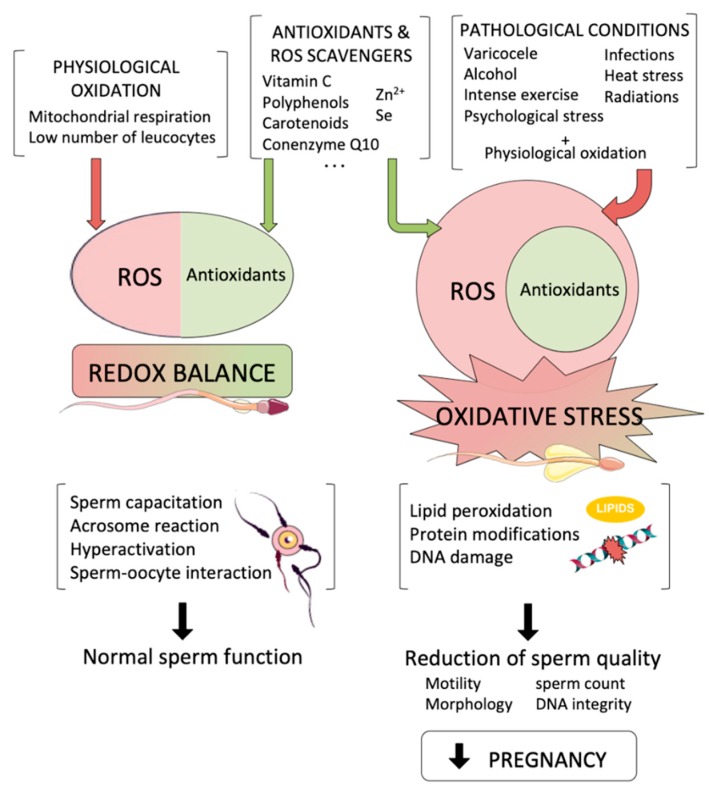
Reactive oxygen species (ROS) are by-products that are necessary for essential sperm functions. On the left, a redox balance is achieved when physiological oxidation is compensated by antioxidants, so that physiological oxidation allows sperm to perform their normal functions. However, when pathological conditions lead to an increase in intracellular ROS levels (right), oxidative stress affects sperm cells causing a reduction in sperm quality that leads to a reduction in pregnancy achievement.

**Figure 2 biology-09-00077-f002:**
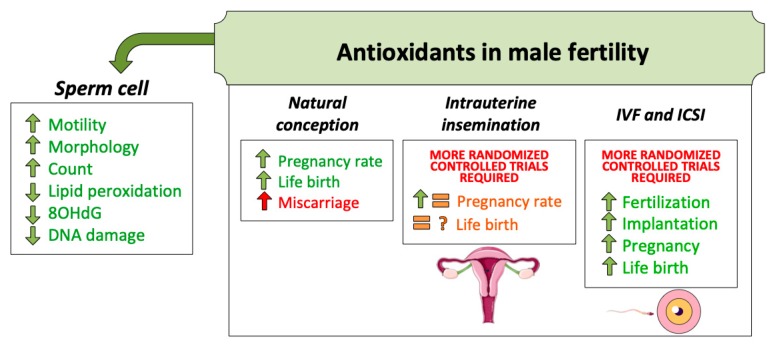
Antioxidant treatments for male fertility have been used for decades. Until the present moment, evidence supports its use to achieve better sperm quality. This, in turn, increases natural conception success. Regarding assisted reproduction techniques (ART), more randomized and controlled trials are necessary to confirm the benefit of oral antioxidant intake.

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
