# Peer review of "Oxidative Stress in Male Infertility: Causes, Effects in Assisted Reproductive Techniques, and Protective Support of Antioxidants"

_biology, 2020, doi:10.3390/biology9040077_

Round 1
Reviewer 1 Report
The review article written by Maynou et al., on the causes and consequences of oxidative stress in male fertility and protective effects of antioxidants, are generally written well. The manuscript has carefully written with a vast amount of information. However, the readability of the manuscript can be better. I have a few suggestions to improve the manuscript as follows,
- The title of the manuscript can be better, as the current title is incomplete
- Line 61, section 2, outlined the causes and consequences of oxidative stress in the male germline, however, the topics are not complied well. The section could be rearranged for clarity. You may also Include the causes of ROS and male infertility by genetic abnormalities, testicular insufficiency etc.
- Line 62, elaborate the endogenous, exogenous sources and generation of ROS. Also, a small paragraph about the physiological role of ROS in seminal plasma could help the reader to know the importance of ROS in the male germline.
The author used sufficient number of references.
Author Response
Authors’ Response to Reviewer 1
The authors thank the reviewer for the time in reviewing the manuscript and their valuable comments. In an effort of improving the readability, the Manuscript has been proofread again. The title has also been revised and modified in an attempt of being more precise.
In order to address the request of the Reviewer, and in consonance with the comments of reviewer 4, we modified the structure of section 2, dividing the global chapter in two chapters, and modifying the structure of the sources of oxidative stress to include the division between endogenous and exogenous sources of ROS. We think, in agreement to the Reviewers, that this structure delivers a more clear message to the reader, achieving a better organization of the sections.
The final structure resulted to be:
- Sources of oxidative stress with potential effects to the male germline
2.1. Endogenous sources of ROS
2.2 Exogenous sources of ROS
- Effects of oxidative stress to sperm components
Following the recommendation of the reviewer, we added a paragraph to show the physiological role of ROS, in section 2.1.
Thank you for reviewing this Manuscript and for the useful comments/improvements.
Reviewer 2 Report
An interesting work regarding analysis of the causes and consequences of oxidative stress in male infertility. The manuscript seems to be comprehensive and well-planned. The authors have provided current information and showed the relationships between oxidative stress and male infertility. All chapters of manuscript had been written in an intelligible way. What is more, presented documentation does not raise any objections. Presented conclusion is understandable and it’s a clear result of the findings gained by the authors.
Author Response
Authors’ Response to Reviewer 2
The authors thank the reviewer for the time in reviewing the manuscript and their evaluation.
Reviewer 3 Report
This review articles discusses the causes and consequences of oxidative stress on male fertility. It also shows some evidence for the roles of anti-oxidants in treatment of infertility.
The manuscript is well written, but needs some improvements:
1-The authors should add a paragraph highlighting ethanol-induced oxidative stress and apoptosis of testicular germ cells. Also, discussing the possible roles of autophagy as protective mechanism stimulated by oxidative stress in animal models of acute and chronic alcoholism will add strength to this review.
2-It is better to show a diagram illustrating how ant-oxidants impact the various options for infertility treatment.
Author Response
Authors’ Response to reviewer 3
The authors thank the reviewer for the time spent in reviewing the manuscript and his/her evaluation to improve its content.
As suggested in the first comment, we included in section Alcohol and tobacco a paragraph related to ethanol-induced oxidative stress that causes an apoptotic and autophagic response at testicular level.
Following the second comment by the Reviewer, we included a diagram illustrating the impact of antioxidants in different infertility treatments.
These changes helped in improving the manuscript, becoming more valuable to the readers.
Thank you very much for the revision.
Reviewer 4 Report
The spermatozoon is a highly specialized cell, whose main function is the transport of the intact male genetic material into the oocyte. During its formation and transit throughout male and female reproductive tracts, sperm cells are internally and externally surrounded by reactive oxygen species (ROS), which are produced from both from endogenous and exogenous sources. While low amounts of ROS are known to be necessary for crucial physiological sperm processes, high levels of those species underlie misbalanced antioxidant-oxidant molecules, generating oxidative stress (OS), which is one of the most damaging factors that affect sperm function and lower male fertility potential.
The present work starts by reviewing the different sources of oxidative stress that affect sperm cells, continues by summarizing he detrimental effects of OS on the male germline, and discusses previous studies addressing the consequences of these detrimental effects on natural pregnancy and assisted reproductive techniques effectiveness. In the last section, they mentioned how antioxidants can counteract the effects of ROS and how sperm fertilizing ability may benefit from these agents.
This is a review of the relationship between male infertility and oxidative stress. Recently, this issue has been a topic in the field of fertility treatment and it is worthwhile to disseminate it as information, but I think it needs a little more correction.
- Line 37: Some report that the frequency of male infertility is close to 50% in some situations. Add more references.
- The paragraph on psychogenic stress from line 133, but no mention of oxidative stress. If there is no evidence please delete.
- The link between electromagnetic waves and oxidative stress, starting at line 147, is now of interest to andrologists worldwide. Please include a bit of evidence data, not just the reference number.
- What is the pollusion of the paragraph from line 162? Please give an example. I imagine you probably want to mention the relation between heavy metal and phthalic acid contamination and oxidative stress.
- Lines 167 to 232: The reader wants to know where ROS is made during spermatogenesis. The sentence from line 168 onwards mentions that, but it is not very detailed. I want you to write a little more detail.
- Also, paragraphs 2.7 to 2.10 summarize the mechanism of ROS generation and the resulting damage to sperm, and how about this as a separate chapter, apart from the sections 2.1 to 2.6?
- If there is data, the effect of OS on natural pregnancy should be described
- Although the only way to reduce OS is to take antioxidants, there is evidence that ROS can be removed by Valicocelectomy or washing semen. If there is room for the number of characters, please state it as well.
- It does not state how often male infertility is caused by OS. This is something that readers want to know. If there are few infertile patients due to OS, there is no need to mention here ...
Author Response
Authors’ response to Reviewer 4
Authors would like to thank the Reviewer for the valuable comments, below we address all the concerns stated point by point:
- We included the information that half of the infertile couples present a male factor, which was indeed lacking. Also, as suggested, we added more references.
- As a recommendation of the reviewer, we deleted the paragraph.
- We rewrote the sub-section including evidences.
- As suggested, we added information about the effects of phthalates on reproductive biology and oxidative stress, thus improving the environmental pollutants section.
- This section has been moved and split for clarity purposes following Reviewer 1’s suggestions. Now this section is numbered as “2.1. Intrinsic sources of ROS”. Following your recommendation, we tried to explain in more detail how ROS is produced intrinsically.
- After considering this point, and also in agreement with Reviewer 1, we restructured this part and treated it as a separate chapter.
- Following your recommendation, we included a short paragraph to reinforce the idea that oxidative stress affects natural pregnancy. Also, the use of antioxidants for natural pregnancy was not clearly differenced as a sub-section in the review. To clearly differentiate natural pregnancy, we added a subsection heading before intrauterine insemination section.
- A subsection about Varicocelectomy has been included as suggested by the Reviewer.
- As stated throughout the review, oxidative stress is the underlying disbalance causing multiple affectations happening in sperm cells, such as an increase in DNA damage, reduction in motility or decrease in sperm count and morphology, to mention some. These parameters are the ones directly related to sperm fertilization potential, and therefore, to male infertility, and are the ones that most studies and clinics analyze. Therefore, ROS and antioxidant capacity are not directly analyzed in male fertility cases, despite possibly being the underlying cause. For this reason, it is rather difficult to attribute a percentage of infertility incidence to oxidative stress.
Thank you very much for the revision
Round 2
Reviewer 3 Report
The authors improved the manuscript vin the current version.
Thanks